# Increased Vascular Endothelial Growth Factor Serum Level and the Role of +936C/T Gene Polymorphism in Chronic Obstructive Pulmonary Disease

**DOI:** 10.3390/medicina57121351

**Published:** 2021-12-10

**Authors:** Ana Florica Chis, Ruxandra-Mioara Râjnoveanu, Milena Adina Man, Doina Adina Todea, Bogdan Augustin Chis, Bogdan Stancu, Ioan Anton Arghir, Teodora Gabriela Alexescu, Carmen Monica Pop

**Affiliations:** 1Department of Pneumology, “Iuliu Haţieganu” University of Medicine and Pharmacy Cluj-Napoca, 8th Victor Babeș Street, 400012 Cluj, Romania; anna_f_rebrean@yahoo.com (A.F.C.); andra_redro@yahoo.com (R.-M.R.); manmilena50@yahoo.com (M.A.M.); doina_adina@yahoo.com (D.A.T.); cpop@umfcluj.ro (C.M.P.); 2“Leon Daniello” Clinical Hospital of Pneumology, 6th Bogdan Petriceicu Hașdeu Street, 400332 Cluj, Romania; 32nd Department of Internal Medicine, “Iuliu Hațieganu” University of Medicine and Pharmacy, 8th Victor Babeș Street, 400012 Cluj, Romania; 42nd Department of General Surgery, “Iuliu Hațieganu” University of Medicine and Pharmacy, 8th Victor Babeș Street, 400012 Cluj, Romania; bstancu7@yahoo.com; 54th Department of Clinical Medical Disciplines II, “Ovidius” University of Medicine and Pharmacy, Mamaia Boulevard, No. 124, 900527 Constanța, Romania; ionut_arghir93@yahoo.com; 65th Departament Internal Medicine, 4th Medical Clinic, University of Medicine and Pharmacy, 400015 Cluj, Romania; teodora.alexescu@gmail.com

**Keywords:** VEGF, chronic obstructive pulmonary disease, rs3025039, ELISA, PCR RFLP

## Abstract

*Background and Objectives*: Chronic obstructive pulmonary disease (COPD) represents a debilitating disease, with rising morbidity and mortality. Vascular endothelial growth factor (VEGF) plays a major role in angiogenesis, vascular permeability, and airway remodeling. The purpose of this study was to investigate the relationship between VEGF serum levels and VEGF +936 C/T gene polymorphism (rs3025039) with COPD, for the first time in a Romanian population. *Materials and Methods*: In total, 120 participants from Transylvania were included in this case-control study. Serum levels of VEGF were determined using an enzyme-linked immune-sorbent assay and rs3025039 was investigated by high molecular weight genomic deoxyribonucleic acid (DNA). Spirometric values, arterial blood gas analysis, and the Six Minute Walk Test (6MWT) outcome were also determined. *Results*: The serum level of VEGF was higher in the COPD group versus controls (*p* < 0.001), with a positive correlation with the 6MWT outcome. No significant difference was observed in the VEGF serum levels between VEGF +936C/T genotypes. There was no difference in the VEGF +936C/T genotype between COPD patients and healthy subjects (chi2 test *p* = 0.92, OR = 1.04, 95%CI = 0.41–2.62), but the presence of the T allele was significantly linked to the presence of COPD (chi2 test *p* = 0.02, OR = 2.36, 95%CI = 1.12–4.97). *Conclusions*: Higher VEGF serum levels were found in moderate and severe COPD and were positively correlated with the distance in the 6MWT. No significant difference was found between CC, CT, and TT genotypes of rs3025039 and the presence of COPD. The presence of the T allele was found to be linked to COPD and also to the degree of airway obstruction.

## 1. Introduction

Chronic obstructive pulmonary disease (COPD) is a multifactorial debilitating disease, characterized by persistent and progressive airflow limitation, with more than 300 million people having COPD worldwide [1]. Exposure to cigarette smoke represents the most cited risk factor, but recent studies have attempted to explain the relatively small percent of smokers that develop COPD. More than one-third of the world’s population is exposed to noxious particles derived from biomass or coal, wood, and charcoal, used as sources of energy or used for cooking (indoor pollution). Occupational exposure to organic and inorganic dust, chemical agents, and fumes, together with high levels of urban air pollution also represent risk factors for COPD (1).

Vascular endothelial growth factor A (VEGF-A), represents a member of a larger family of growth factors that have different expression patterns and biological functions: VEGF-B, VEGF-C, VEGF-D, and placental growth factor (PLGF) [2]. VEGF is a soluble heparin-binding glycoprotein involved in endothelial cell proliferation, migration, and survival through Flt-1 and Flk-1 (receptors in the membrane that are bound) [3]. The human VEGF gene is located on chromosome 6p21.3 and is composed of 8 exons and 7 introns [4]. This cytokine has potent angiogenic properties, modulates thrombogenicity, and enhances vascular permeability [5]. VEGF has been studied in various pathologies, and in the field of COPD, studies have shown that VEGF deficiency, or its ability to signal, augments oxidant injury and tissue destruction and is involved in the pathogenesis of severe pulmonary emphysema, through apoptotic and oxidative stress mechanisms [6]. In contrast, VEGF levels in induced sputum and blood serum were found to be elevated in asymptomatic smokers and in COPD smokers without alveolar destruction [7].

In the past decades, researchers have tried to define what susceptibility means in terms of COPD, with interesting results in the field of gene polymorphisms. Results from genome-wide association studies identified new loci in the alfa-1-antitrypsin gene, tumor necrosis factor genes, microsomal epoxide hydrolase gene, glutathione S-transferases gene, and others that might contribute to the pathogenesis of COPD but require further validation [8]. Although current results are promising, the presence of interethnic variation suggests that ethnicity-specific reference intervals may be necessary [9].

Single nucleotide polymorphisms (SNPs) represent the most common type of genetic variation among individuals and occur when a single nucleotide in the genome (or other shared sequence) is replaced with another, between members of a species or paired chromosomes in an individual. Genetic variations in the VEGF gene might lead to high-level expression of VEGF. Overexpression of VEGF in the lungs increases the pulmonary vascular permeability but also augments neovascularization, contributing to the repair of endothelial injuries [10]. Several VEGF SNPs, −2578C>A, −460C>T, +936C>T, +405C>G, −1154G>A, −14625T>C, and −583T>C, have been described in relation to pulmonary diseases, especially COPD, asthma, and lung cancer [11,12].

The aim of this study was to investigate the relationship between VEGF serum levels and VEGF +936 C/T gene polymorphism (rs3025039) with COPD and with several characteristics of these patients, in a group of patients from the Transylvania region. To our knowledge, this is the first study to investigate the relationship between SNP and COPD in a Romanian population.

## 2. Materials and Methods

*Study population.* A case-control observational study was conducted, authorized by Research Ethics Committee of “Iuliu Hațieganu” University of Medicine and Pharmacy, Cluj-Napoca, no.298/29.06.2016. All the participants (60 patients with COPD and 60 control healthy subjects, all from the Transylvania region of Romania) signed the written informed consent. The two groups were comparable in terms of age, sex, body mass index (BMI), smoking history, and the environment of origin (urban/rural). The diagnosis and classification of COPD was based on 2017 Global Initiative for Obstructive Lung Disease (GOLD) guideline [13]. The exclusion criteria were: younger and elders (age under 40 years or over 90), known diagnosis of pulmonary disease (asthma or obstructive sleep apnea, enzymatic deficiencies, interstitial tissue disorders, pulmonary hypertension, or acute diseases like pneumonia), cardiovascular diseases (heart failure, stroke, peripheral arterial or venous disease or injuries, recent revascularization), chronic medications (oral anticoagulants, corticosteroid treatment), recent surgery, kidney disease, neoplasia, or autoimmune disorders. In both groups (patients and controls), the serum level of VEGF was determined, and VEGF +936 C/T gene polymorphism was investigated. For the COPD patients, named the study group, the assessment methods included evaluation of the smoking status, body mass index (BMI), spirometry, partial tension of oxygen (PaO_2_) carbon dioxide (PaCO_2_), derived from arterial blood gas analysis, and six-minute walk test (6MWT) according to the standard protocol of the 2002 American Thoracic Society (ATS) Guidelines [14]. All the patients were divided into active smokers (ASs), non-smokers (NSs), and former smokers (FSs), while Pack-Year Index (PYI) was used to quantify smoking history. Functional lung explorations were performed according to the ATS and European Respiratory Society (ERS) guidelines [15,16].

*Blood sampling and analysis.* Blood samples (3 mL) were taken in the first or second day of admission, early in the morning, after overnight fasting, from the antecubital vein, and were processed (centrifugation at 3000 rpm for 10 min) within 30 min of collection. Serum was aliquoted and stored at −20 °C until analysis. The measurements were double batched tested. The concentrations of VEGF were measured using commercial, enzyme-linked immune-sorbent assay technology (Human VEGF PicokineTM ELISA Kit, Boster biological technology Co., Ltd., 3942 B Valley Ave, Pleaseantonm, CA 94566, USA, Catalog Number EK0539). The VEGF range of detection was 31.2 to 2000 pg/mL, and the sensitivity of the assay was ˂1 pg/mL. The blood oxygen saturation (SpO_2_), PaO_2_, and PaCO_2_ were determined by peripheric arterial puncture, in sitting position, breathing room air, early in the morning. Respiratory failure was considered if PaO_2_ ˂ 60 mmHg was observed [17].

*Genotyping conditions.* Blood samples were collected from all participants in EDTA (ethylenediaminotetraacestic acid) coated tubes, after overnight fasting. DNA was extracted using 0.3 mL of blood with a DNA Purification Commercial Kit (Wizard Genomic produced in Fitchburg, Wisconsin, United States of America by Promega Corporation). Genotyping was done by Polymerase Chain Reaction Restriction Fragment Length Polymorphism (PCR RFLP). Specific DNA amplification was obtained from 0.1 µg of genomic purified DNA. The total amount was amplified in 25 μL of reaction mixture. The reaction mixture used contained 12.5 μL of ready-to-use solution (Taq DNA polymerase—PCR Mastermix along with deoxynucleotide triphosphates, MgCl_2_, and reaction buffers), 7.5 μL of free nuclease water, 1 μL of bovine serum albumin, 1 μL of each primer, and 1 μL of water suspended DNA. The investigated polymorphism was VEGF +936C>T, also known as rs 3025039 and the primers were: F3′-AAGGAAGAGGAGACTCTGCGCAGAGC-5′ and R5′-TAAATGTATGTATGTGGGTGGGTGTGTCTACAGG-3′, with NLA III as the restriction enzyme. Enzyme digestion was done on the amplification products (Fermentas; Thermo Fisher Scientific, Waltham, MA, USA) followed by electrophoresis analysis on agarose gel (MetaPhor^®^; FMC BioProducts, Rockland, ME, USA) and detection of genotypes: CC (208 bp), CT (208, 122, 86 bp) and TT (122, 86 bp). The C allele was considered the normal wild type and the T allele as the variant.

*Statistical analysis.* The statistical analyses were performed using Microsoft Excel from Microsoft Office 2016 software (Microsoft Co, Redmond, WA, USA), CDC Epi Info (version 7; CDC, Atlanta, GA, USA), and IBM SPSS (version 20.0; SPSS, Chicago, IL, USA). Student’s *t*-test and ANOVA test were used for differences between groups. The goodness-of-fit test was performed with Hardy–Weinberg equilibrium for the genotype distribution. The assessment of VEGF serum levels between groups included the Kolmogorov–Smirnov test, followed by the Mann–Whitney U Test for independent samples and the Kruskal–Wallis test for multiple group comparison. Pearson χ2 and Fisher’s exact tests were used to analyze the genotype frequencies. Logistic regression and Fisher’s test with odds ratios (ORs-95% confidence intervals) were used for risk factor assessment. *p* < 0.05 was considered statistically significant.

## 3. Results

The demographic characteristics of the study and control group are presented in Table 1. No difference was observed between the two groups in terms of age, sex, and smoking history. The COPD patients had a higher PYI (29.5 ± 18.2 vs. 23.4 ± 11.3 in healthy subjects), with no statistically significant difference between the two groups (*p* = 0.06).

VEGF levels were higher in patients’ serum compared to controls (103.91 ± 66.48 vs. 4.16 ± 11.63, *p* < 0.001). In the control group, no correlation was found between VEGF serum levels and smoking history (*p* = 0.45). When assessing correlations between VEGF serum levels and clinical and paraclinical parameters for the COPD patients, intragroup analysis revealed no correlation between VEGF serum level and age, smoking history, severity of the disease, SpO_2_, PaO_2_, or PaCO_2_. VEGF serum level correlated positively with the outcome in the 6MWT (Table 2).

For both groups, the Hardy–Weinberg equilibrium in genotype frequencies was not present (chi2 test *p* < 0.001 for COPD patients and chi2 test *p* < 0.001 for healthy subjects). The distribution of genotypes and allele is presented in Table 3.

The T allele was the minor allele and was identified in 20.83% of the patients and 10% in the control group. The TT genotype was not present in the control group, and we found no statistically significant relationship between genotype and the presence of COPD (CT vs. CC, *p* = 0.92, OR = 1.04, 95% CI = 0.41–2.62), but a significant difference was revealed when the T allele was considered (*p* = 0.02, OR = 2.368, 95% CI = 1.12–4.97).

No significant difference was observed between VEGF level and alleles or genotype inside the groups, and there is only a global difference, induced by the low level of VEGF in the control group, but without statistical significance (Table 4).

When evaluating the influence of the genotype of rs3025039 in inducing modifications in follow-up/prognostic parameters in COPD patients, no significant difference was observed in terms of FEV1s, PaO_2_, PaCO_2_, and distance in the 6MWT (Table 5), with a *p* value > 0.05 in all cases. Similarly, the t student test was applied in order to identify differences in these parameters according to the presence of the T allele. No difference was found between groups for the PaO_2_ (*p* = 0.79), PaCO_2_ (*p* = 0.85), or the result of the 6MWT (*p* = 0.33), but a statistically significant difference was observed when FEV1s was analyzed. The mean value of FEV1s in patients with the T allele was 37.92 ± 11.4, which was significantly lower than those with the C allele: 45.49 ± 16.43, with a *p* value of 0.01.

## 4. Discussion

In terms of prevalence, mortality, morbidity, and healthcare costs, COPD represents a real burden, worldwide. Studies of the past two decades revealed important pathogenetic aspects regarding lung tissue remodeling in COPD patients: alterations of the mucosal tissue, fibrosis, inflammation (both local and systemic), vascular remodeling, and angiogenesis in the lungs [18,19,20]. Angiogenesis was first described by Leonardo DaVinci, by studying heart and lung circulation [21], and intensive studies have established the major role of angiogenesis in chronic inflammatory processes like asthma [22], tumor development, as well as early cancer pathogenesis, including lung cancer [23,24]. In the process of angiogenesis, several angiogenic growth factors are involved, including VEGF, angiogenin, fibroblast growth factor, and placenta growth factor, with VEGF having the most important role in remodeling of blood vessels [25]. VEGF, secreted by endothelial cells, macrophages, and stromal and malignant cells [26], stimulates and activates proteolytic enzymes and induces degradation of the extracellular matrix [27]. This leads to the organization of endothelial cells in tubular structures, through proliferation and migration [28]. In 2005, Kranenburg et al. found an association between the expression of VEGF in bronchiolar and alveolar regions (bronchial, bronchiolar, and alveolar epithelium, airways and vascular smooth muscle cells) of COPD patients [29]. These findings have led to the hypothesis that the symptoms in COPD could be controlled with pharmacological drugs, e.g., inhibitors of VEGF and its receptors [30]. In animal models (rats), Sunitinib treatment was associated with less angiogenesis in small-airway remodeling and lower levels of VEGF and its receptors [25]. A present subject of interest is finding new angiogenic inhibitors that are useful in treating lung diseases with overexpression of angiogenic factors [31]. Valipour et al. reported the results of a comparison between circulating VEGF levels in exacerbated COPD, stable COPD, and healthy subjects, and found an increased value of VEGF serum levels for both stable and exacerbated COPD patients, with a positive correlation between VEGF level and FEV1s, but no correlation with age, BMI, PYI, PaO_2_, or PaCO_2_ [32]. In 2014, Hosseini et al. published a paper that reveals that VEGF serum levels are higher in COPD patients, regardless of the smoking status, and this value proportionally increases with the severity of the disease [7]. Conversely, in 2015, Boeck et al. found that although VEGF serum levels are higher in COPD patients (including those with cardiac diseases and diabetes), the values were not associated with clinically significant outcomes (FEV1s, result in 6MWT, BODE index) in COPD [33]. The results of our study are consistent with the literature, since we found significantly higher levels of serum VEGF in the COPD group versus healthy subjects, without any correlation with age, smoking history, or arterial blood gases. Moreover, contrary to previous studies, we found a positive correlation of VEGF serum levels with the results of the 6MWT (meters). We consider this to be important, since the 6MWT has been widely recognized as a useful tool in appreciating the quality of life and the prognostic of COPD patients [34]. Imagistics, such as computed tomography of the thorax, might be able to quantify the extent of emphysema and compare it to the levels of VEGF, since patients with increased endothelial dysfunction have reduced 6 min walk test (6MWT) results and a worse overall prognosis [35].

In 2014, Wu et al. published a review that characterized the evidence that gene polymorphisms contribute to the etiology of COPD and also explored the potential relationship between these gene polymorphisms and certain clinical characteristics. Several gene polymorphisms were investigated, such as Alpha 1-antitrypsin gene, Tumor necrosis factor genes, Microsomal epoxide hydrolase gene, Glutathione S-transferases genes, Transforming growth factor-beta(1) gene, and IL6 and IL8 gene, with VEGF polymorphisms not being among them [36]. Two years later, in 2017, Yuang et al. summarized the relationship between COPD and a genome-wide association study, epigenetics, apoptosis, gene polymorphisms, and genes, without any information on VEGF polymorphisms and its relationship with COPD [37]. However, over the last 3 years, several studies have tried to describe the role of VEGF SNPs in COPD susceptibility, characteristics, prognosis, and response to drug therapies. In a study on the Chinese population of Hainan province, the authors found, contrary to previous data, that several alleles (C allele from rs3025030 and G allele from rs3025033), haplotypes (GC of VEGFA), and rs9296092 represent risk factors for COPD [38]. In 2017, Yu and colleagues found no statistically significant difference in the genotype and allelic frequencies of VEGF rs699947 and rs1570360 between COPD patients and control groups [39]. We investigated the relationship between VEGF rs3025039 and COPD as we found a wide range of studies that focused on this polymorphism and its relationship with various diseases, like pre-eclampsia [40], extracranial internal carotid artery stenosis and ischemic stroke [41], and bladder cancer [42], but only one study that described its involvement in COPD. Baz-Davila et al. investigated the potential role of several SNPs in the susceptibility and progression of COPD in a Spanish population and found no association between rs3020539 and COPD [43]. Our results are similar to these findings, as the CC, CT, and CT genotype were not associated with the disease, but in our Romanian cohort of COPD patients, we found that the T allele represents a risk factor for COPD (increased risk of 2.36 for the population with the T allele) and is also associated with a decrease in lung function (FEV1s).

## 5. Conclusions

VEGF serum levels were higher in moderate and severe COPD patients and positively correlated with the outcome of the 6MWT. No significant difference was found between the CC, CT, and TT genotypes of rs3025039 and the presence of COPD. The presence of the T allele increases the risk of COPD by 2.36 and is also linked to the degree of airway obstruction. The functional parameters (FEV1, FVC, SpO2, PaO2, PaCO2) are not influenced by the presence of the CC/CT or TT genotype of rs3025039 polymorphism.

## Figures and Tables

**Table 1 medicina-57-01351-t001:** Demographic characteristics and clinical parameters of participants.

	Cases (*n* = 60)		Controls (*n* = 60)	*p* Value
Age	66.2 ± 9.3		65.1 ± 10.3	0.53 ^a^
Gender Male/Female	54/6		56/4	0.48 ^b^
BMI	29.4 ± 6.3		28 ± 5.1	0.55 ^a^
FEV1s (*n*, %)	<30%	13 (21.6%)		
	30–50%	28 (46.6%)		
	≥50%	19 (31.6%)		
PaO_2_	70.5 ± 18.6			
PaCO_2_	44.7 ± 9.56			
SpO_2_	90.8 ± 5.92			
6MWT (m)	303.9 ± 97.7			

*n* = number of individuals; SD = standard deviation, the values are presented as mean ± SD, BMI = Body Mass Index, FEV1s = Forced Expiratory Volume in 1st second, PaO_2_ = partial tension of oxygen; PaCO_2_ = partial tension of carbon dioxide, SpO_2_ = peripheral blood oxygen saturation, 6MWT = Six Minute Walk Test, ^a^ = based on the Student’s *t*-test; ^b^ = based on Chi-square test.

**Table 2 medicina-57-01351-t002:** Correlation between the VEGF serum level and the characteristics of the COPD patients.

		Age	PYI	6MWT Distance	FEV1s (%)	SpO_2_	PaO_2_	PaCO_2_
VEGF serum levels COPD group	Pearson Correlation	−0.209	0.009	0.286	−0.075	0.020	0.004	−0.098
	Sig. (2-tailed)	0.109	0.948	0.027	0.567	0.882	0.973	0.457

VEGF = vascular endothelial growth factor, COPD: chronic obstructive pulmonary disease. PYI = pack year index, 6MWT = Six Minute Walk Test, PaO_2_ = partial tension of oxygen; PaCO_2_ = partial tension of carbon dioxide, SpO_2_ = peripheral blood oxygen saturation.

**Table 3 medicina-57-01351-t003:** Distribution of genotypes and alleles in the COPD group and control group.

		COPD Patients		Healthy Subjects	
		Number	Percent	Number	Percent
Genotypes	CC	42	70.0%	48	80.0%
	CT	11	18.3%	12	20.0%
	TT	7	11.6%	0	0.0%
Alleles	C	95	76.16%	108	90.0%
	T	25	20.83%	12	10.0%

COPD = Chronic Obstructive Pulmonary Disease.

**Table 4 medicina-57-01351-t004:** Analysis of the VEGF serum levels by genotypes and alleles.

		COPD Patients		Healthy Subjects		Global	
		Mean	Standard Deviation	Mean	Standard Deviation	Mean	Standard Deviation
Genotypes	CC	104.476	66.265	3.771	11.258	50.76	68.134
	CT	89.63	69.068	5.75	13.471	45.87	63.991
	TT	123	68.595	-		123	68.595
Probability	0.413 **			0.573 *		0.014 **	
Alleles	C	102.758	66.036	3.991	11.419	50.212	67.373
	T	108.32	68.015	5.75	13.471	75.054	74.223
Probability	0.831 *			0.594 *		0.026 *	

*n*: number of patients, COPD = Chronic Obstructive Pulmonary Disease, * Mann–Whitney U Test, ** Kruskal–Wallis Test.

**Table 5 medicina-57-01351-t005:** Impact of CC, CT, and TT genotypes of VEGF rs3025039 polymorphism on paraclinical prognostic and follow-up parameters.

	Genotypes					
	CC		CT		TT	
	Mean	Standard Deviation	Mean	Standard Deviation	Mean	Standard Deviation
PaO_2_	71.317	20.982	72.282	12.225	63.400	10.511
PaCO_2_	44.988	10.530	42.482	6.892	46.700	7.029
FEV1s (%)	45.495	15.584	43.094	12.659	41.486	22.422
FVC (%)	77.102	14.701	75.927	7.340	75.214	8.077
6MWT distance	312.976	99.671	279.545	81.623	287.857	114.231

FEV1s = Forced Expiratory Volume in 1st second, FVC = Forced Vital Capacity, PaO_2_ = partial tension of oxygen; PaCO_2_ = partial tension of carbon dioxide, 6MWT = Six Minute Walk Test.

## Data Availability

The data sets are available upon request form the first authors.

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
