# Peer review of "Increased Vascular Endothelial Growth Factor Serum Level and the Role of +936C/T Gene Polymorphism in Chronic Obstructive Pulmonary Disease"

_medicina, 2021, doi:10.3390/medicina57121351_

Round 1
Reviewer 1 Report
Introduction.
i) Cigarette smoke is correctly presented as "the most cited risk factor" for COPD, but occupational risk factor (e.g. for welders, miners, melters / casters...) deserve to be considered too.
ii) "new loci (...) might contribute to (instead of "define") the pathogenesis of COPD".
Matherials and methods
Cases and controls have been studied in consideration of the serum levels of VEGF and of the related polymorphism, not of their smoking habits, and no attention has been given to possible occupational risk factors unbalanced in the two groups. Matching criteria are insufficiently described.
Author Response
Thank you for your valuable sugestions.
Q1- paragraph added
Q2- paragraph changed
We rephrased and extended the Introduction and Material and methods.
We rephrased the Tables 3 and 4.
Thank you for your time and sugestions.
Reviewer 2 Report
The paper is of certain interest. The only strong finding is the significantly higher circulating level of VEGF in COPD patients compared with Controls, this should acknowledged in the title. Possible clinical implications should also be discussed.
Author Response
Thank you for your valuable sugestions.
Q1- title changed
Q2- paragraph changed
We rephrased and extended the Introduction , Material and methods and Discussions.
We rephrased the Tables 3 and 4.
Thank you for your time and sugestions.
Round 2
Reviewer 1 Report
A characterization of cases and controls on the basis of their occupations and corresponding profiles of occupational exposures to dusts, NOx, mists .... would have been useful. These aspects would be considered for ongoing research.